# Perturbed N-glycosylation of *Halobacterium salinarum* archaellum filaments leads to filament bundling and compromised cell motility

Shahar Sofer[1,3], Zlata Vershinin [1,3], Leen Mashni[1,3], Ran Zalk [2], Anat Shahar[2], Jerry Eichler [1] & Iris Grossman-Haham [1,2] ✉

The swimming device of archaea—the archaellum—presents asparagine (N)-linked glycans. While N-glycosylation serves numerous roles in archaea, including enabling their survival in extreme environments, how this post-translational modification contributes to cell motility remains under-explored. Here, we report the cryo-EM structure of archaellum filaments from the haloarchaeon *Halobacterium salinarum*, where archaellins, the building blocks of the archaellum, are N-glycosylated, and the N-glycosylation pathway is well-resolved. We further determined structures of archaellum filaments from two N-glycosylation mutant strains that generate truncated glycans and analyzed their motility. While cells from the parent strain exhibited unidirectional motility, the N-glycosylation mutant strain cells swam in ever-changing directions within a limited area. Although these mutant strain cells presented archaellum filaments that were highly similar in architecture to those of the parent strain, N-linked glycan truncation greatly affected interactions between archaellum filaments, leading to dramatic clustering of both isolated and cell-attached filaments. We propose that the N-linked tetrasaccharides decorating archaellins act as physical spacers that minimize the archaellum filament aggregation that limits cell motility.

The archaellum (or the archaeal flagellum) is a specialized organelle that enables cell motility in archaea, serving as the functional equivalent of its bacterial counterpart. As with the bacterial flagellum, a motor anchored in the membrane rotates an extracellular archaellum filament, resulting in cell propulsion[1,2]. Archaellins correspond to the basic building blocks of the archaellum filament[3], where they are organized into a helical array[4]. Archaellins were, moreover, among the first archaeal proteins shown to be asparagine (N)-glycosylated[5]. In archaea, N-glycosylation has been assigned various roles, including contributing to the ability of these microorganisms to survive in the extreme environments they can inhabit,

ensuring effective cell mating and other cell–cell interactions, affecting protein complex formation, and impacting viral binding and infection[6–18]. Still, it remains largely unknown how N-linked glycans fulfill these roles. At the same time, despite the almost universal appearance of N-glycosylation in archaea, the composition and architecture of N-linked glycans decorating archaeal glycoproteins, including archaellins, have only been addressed in a few tens of species, with the pathways used to assemble these glycans having been fully or partially delineated in but a handful[9,10,18].

*Halobacterium salinarum* (*Hbt. salinarum*), halophilic archaea that grow in NaCl concentrations near or at saturation, harbor

[1]Department of Life Sciences, Ben-Gurion University of the Negev, Beer Sheva, Israel. [2]The Ilse Katz Institute for Nanoscale Science and Technology, Ben-Gurion University of the Negev, Beer Sheva, Israel. [3]These authors contributed equally: Shahar Sofer, Zlata Vershinin, Leen Mashni. ✉e-mail: irisgh@bgu.ac.il

archaella comprising several filaments, each rotated by its own motor and assembled from N-glycosylated archaellins[5,18–20]. Initially isolated from salted fish over a century ago[21], *Hbt. salinarum* first drew general attention with the 1971 discovery of bacteriorhodopsin, the light-driven proton pump isolated from purple membranes of this organism[22]. Since, studies of *Hbt. salinarum* have led to numerous paradigm-shifting discoveries, including the first example of protein glycosylation outside Eukarya. Specifically, the surface (S)-layer glycoprotein that forms the S-layer surrounding the cell was shown to be both N- and O-glycosylated[23–25]. Soon after, archaellins were also shown to be N-glycosylated[5]. More recent NMR and mass spectrometry efforts have defined the composition and architecture of an N-linked tetrasaccharide decorating these glycoproteins as comprising a glucose, a glucuronic acid, a sulfated iduronic acid, and a terminal sulfated glucuronic acid[26,27]. Furthermore, those archaellin asparagine residues modified by this glycan have been described, as have enzymes involved in *Hbt. salinarum* N-linked tetrasaccharide assembly[28–30]. Specifically, Agl28, Agl25, Agl26, and Agl27 are glycosyltransferases, respectively, responsible for adding the first, second, third, and fourth tetrasaccharide sugars to the dolichol phosphate (DolP) carrier upon which the glycan is assembled. Agl29 is responsible for (or contributes to) flipping the DolP-bound tetrasaccharide across the plasma membrane, at which point the oligosaccharyltransferase AglB delivers the tetrasaccharide to selected asparagine residues in target proteins, including archaellins[28–30]. In addition to these advances in our understanding of *Hbt. salinarum* N-glycosylation, the importance of this post-translational modification for *Hbt. salinarum* physiology has also been recognized. For instance, a link between proper N-glycosylation and cell motility was demonstrated in strains lacking AglB or those glycosyltransferases responsible for N-linked tetrasaccharide assembly. Plate motility assays showed these mutant strains to be less motile than the parent strain, with the impact on cell motility being more pronounced as the number of sugars comprising the N-linked tetrasaccharide was decreased or in the absence of this glycan[28–30]. Indeed, a similar effect of N-linked glycan truncation, or the absence thereof, has been seen in other archaea, with the extent of compromised motility differing among different mutants and species[31–35]. The explanation offered in some of these studies was that compromised N-glycosylation affects archaellum assembly and structure, and by extension, function, reflected as compromised swimming ability.

To date, archaellum filament structures from several archaea have been described. Earlier work addressing *Hbt. salinarum* archaellum structure relied on negative-stain electron microscopy (EM) to determine the helical symmetry of archaellins within the archaellum filament[4,36]. However, as these structural studies were limited in resolution, they could not resolve the contributions of N-glycosylation to archaellum structure nor to its motility. More recent high-resolution structures of archaella from several other species provided indications of glycosylation at the archaellum surface[37–42]. Yet, since little is known of N-glycosylation in the vast majority of archaeal species, most of these archaellum structural models did not include the native N-linked glycans decorating their archaellins nor consider the contribution of N-glycosylation to archaellum structure and function.

In the present study, cryo-EM and helical reconstruction were used to determine the high-resolution structure of the archaellum filament from *Hbt. salinarum*, as well as the first archaellum structures from N-glycosylation mutant strains. In addition, the motility of individual cells from these strains was quantitatively assessed. Comparative analyses of the structures obtained and the swimming behavior of cells from the parent and mutant strains provide new insight into how archaellin N-glycosylation affects archaellum interfilament packing and swimming motility.

# Results

## High-resolution cryo-EM structure of the *Hbt. salinarum* archaellum filament

The structure of the *Hbt. salinarum* archaellum filament was determined from archaella isolated from 4 M NaCl-containing medium and diluted into 1 M NaCl-containing buffer. Such dilution of the salt concentration, which did not affect filament intactness (Supplementary Fig. 1a), was necessary to reduce noise in cryo-EM micrographs. To reconstruct a 3D map of the archaellum filament from cryo-EM micrographs, we performed helical reconstruction, using the twist (108°) and helical rise (5.4 Å) values determined previously from negative-stained EM micrographs as initial parameters[36]. The twist and helical rise were refined to values very close to the initial values, yielding a map at an average resolution of 3.4 Å (Supplementary Fig. 1b–d). Since *Hbt. salinarum* encodes five archaellins, i.e., ArlA1, ArlA2, ArlB1, ArlB2, and ArlB3 (formerly FlaA1, FlaA2, FlaB1, FlaB2, and FlaB3)[43,44] (Supplementary Fig. 2), all of which were detected in spent medium of the culture[29,45], and their arrangement within archaellum filaments is unknown, we further refined the cryo-EM map without applying symmetry in an attempt to resolve the positions of these archaellins within the filament, as done previously with reconstruction of the *Methanocaldococcus villosus* archaellum[37,46]. Symmetry-free refinement improved the overall resolution map to 3.2 Å (Fig. 1a; Supplementary Figs. 1e f, and 3; Supplementary Table 1), and revealed differences in density among archaellin subunits. Nonetheless, we were unable to identify features that would allow us to unambiguously assign specific archaellins into the density (Supplementary Fig. 4), perhaps because those regions that distinguish each archaellin are few, short, and mostly predicted to lack defined secondary structure (Supplementary Fig. 2), or because the five archaellins are not organized in a repeating pattern, but are instead randomly distributed within the basic repeating unit used to generate the filament[47]. Moreover, those regions unique to each archaellin face the solution and hence do not participate in inter-molecular interactions (Supplementary Fig. 5), preventing the prediction of their arrangement based on surface complementarity. Consequently, we built a model into the central region of the cryo-EM map (Supplementary Fig. 1e) comprising 26 archaellin subunits that share a consensus sequence, in which identical residues among the five archaellins were explicitly modeled, and with variable positions usually being modeled as alanine residues (Fig. 1b; Supplementary Table 2; "Methods").

Like archaellins in other species, the *Hbt. salinarum* archaellin is tadpole-shaped, consisting of a 45 residue-long N-terminal α-helix and a C-terminal globular domain organized into an eight-stranded antiparallel β-sandwich (Fig. 1c)[37–42,48]. The two domains are connected by a short three amino acid linker, which appears to be rigid, adopting a similar conformation in all archaellins (Supplementary Fig. 6). For each archaellin, the density of the cryo-EM map enabled the building of a model starting at glutamine 13, indicating that the preceding 12 amino acids serve as a signal peptide, which is removed from the protein before its assembly into an archaellum filament (Supplementary Fig. 2), as predicted[28,49].

The overall packing of archaellins into *Hbt. salinarum* archaellum filaments resembles that seen in other archaea[37–42,48]. The globular domains of the archaellins face outwards and constitute the hydrophilic surface of the filament, which is mostly negatively charged (Supplementary Fig. 7), while the α-helices, which are largely hydrophobic, bundle to form the filament core (Fig. 1b). Each globular domain forms inter-molecular interactions (primarily polar) with the globular domains of six adjacent subunits. Lateral interactions between the globular domains create three left-handed helical strands, vertical interactions form seven right-handed helical strands, and small interfaces at the edges of the globular domains create a four-start helix (Fig. 1d). Ten-fold symmetry is observed from the axial direction due to the presence of ten subunits in a full turn of the left-handed helical

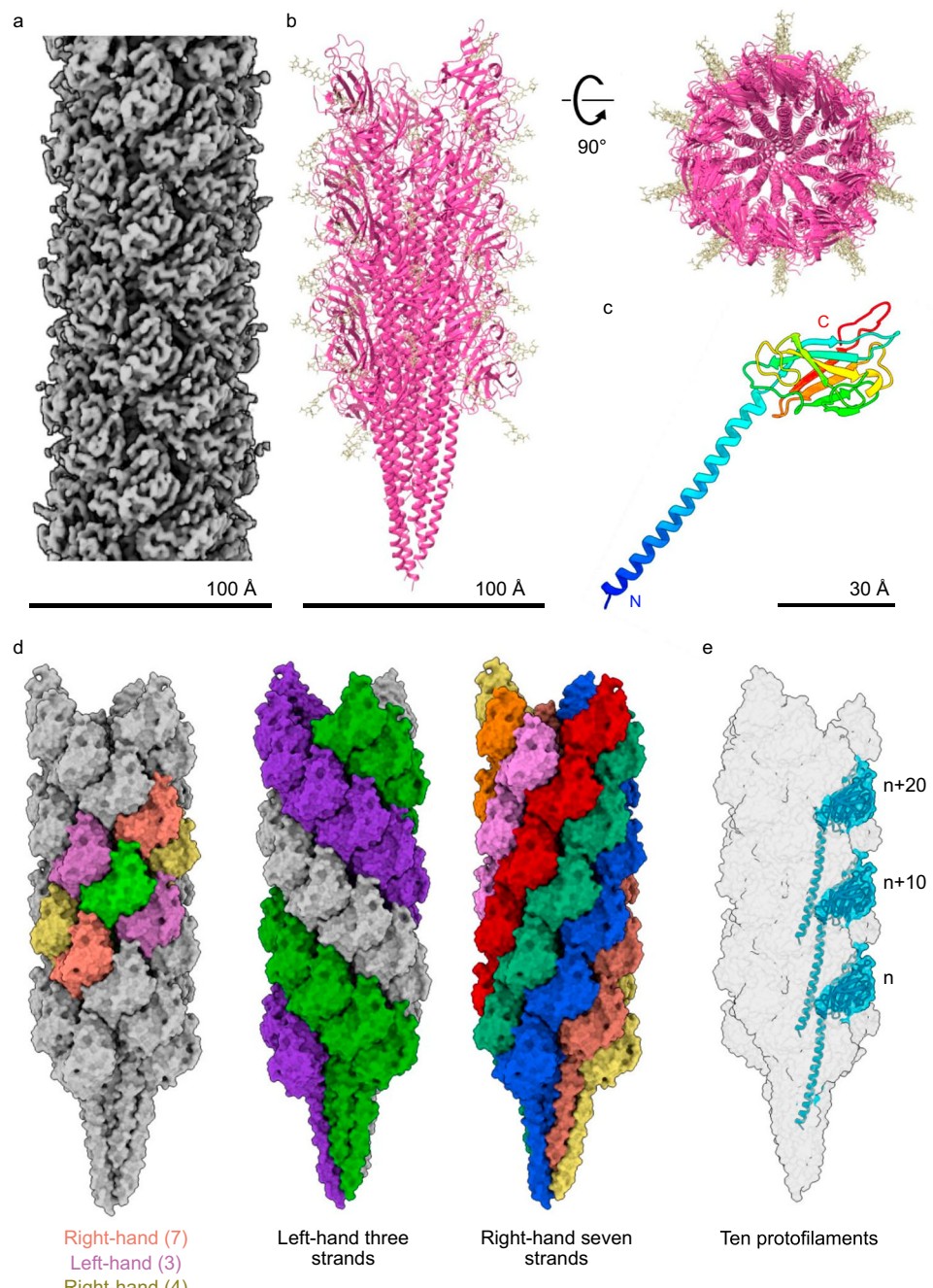

**Fig. 1 | High-resolution cryo-EM structure of the *Hbt. salinarum* archaellum filament. a** Cryo-EM map of the *Hbt. salinarum* archaellum filament. See Supplementary Fig. 1 for more details. **b** Atomic model of an *Hbt. salinarum* archaellum filament, including 26 archaellin subunits. Proteins are shown in pink and N-linked tetrasaccharides are in brown. **c** Model of one archaellin subunit colored from the N-terminus to the C-terminus using a rainbow color scheme. N-linked tetrasaccharides were removed for clarity. **d** Surface presentations of an archaellum filament model with inter-subunit interactions mediated by archaellin globular domains shown. The green archaellin in the left-most representation serves to describe the interactions of each archaellin in the filament and its six neighbors. Lateral interactions (formed with subunits colored pink in the left-most representation) form three left-handed strands (shown in the center representation, with each helical strand differently colored). Vertical interactions (formed with subunits colored orange in the left-most representation) form seven right-handed strands (shown in the right-most representation, with each helical strand differently colored). Interactions with subunits colored yellow form four right-handed strands (not shown). **e** Transparent surface view with subunits (colored blue) that form one protofilament.

strands (Fig. 1b; Supplementary Fig. 6). Thus, the $n$ and $n+10$ subunits along the three-start helix coincide into one protofilament through contacts between the helical domains and the inner part of the globular domains (Fig. 1e). In certain archaellum filament structures, one of the ten subunits adopted a different conformation, as compared to the other nine, due to supercoiling of the filament into a corkscrew-like structure[42,48]. However, in our *Hbt. salinarum* archaellum structure,

there was no difference in the configuration of the linker region connecting the N-terminal helix and the C-terminal globular domain or in archaellin conformation among the ten subunits (Supplementary Fig. 6), presumably because the conditions under which the cryo-EM data were collected (i.e., in 1 M NaCl) did not enable the archaellum filaments to supercoil, but rather to straighten[36] (Supplementary Fig. 1a).

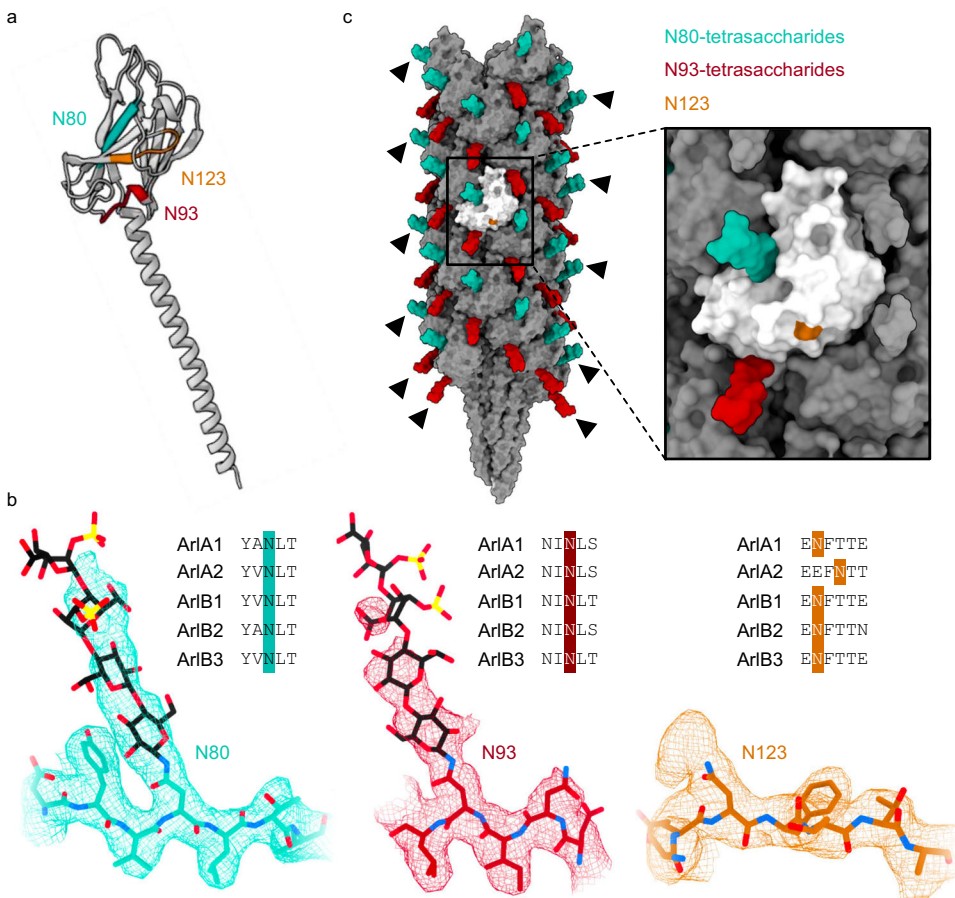

**Fig. 2 | N-glycosylation of archaellins in the *Hbt. salinarum* archaellum.**
**a** Cartoon presentation of an archaellin, showing the positions of the three N-glycosylation sites and neighboring residues in different colors. **b** For each colored region highlighted in (**a**), the alignment of the five archaellin sequences (top right), the cryo-EM map (displayed as a mesh), and our consensus model (in stick presentation) of the archaellin and N-linked tetrasaccharide (N80 and N93) are presented. N80, N93, and N123 are colored cyan, red, and orange, respectively, as are other residues in the surroundings of each asparagine residue. Tetrasaccharide sugar carbons are colored black. **c** Surface presentation of the archaellum filament model with the tetrasaccharides linked to N80 and N93 colored cyan and red, respectively. Arrowheads indicate tetrasaccharides, which in this view, clearly protrude from the protein surface, instead of forming interactions with neighboring archaellins. The box on the right provides a closer view of one archaellin colored white with its tetrasaccharides linked to N80 and N93 colored cyan and red, respectively. In addition, the position of N123 is indicated in orange (the N-linked tetrasaccharide was not modeled at this position because only residual density was observed in the cryo-EM map), showing it is not found at the interface with neighboring archaellins, which are colored gray.

## N-glycosylation of archaellins in the *Hbt. salinarum* archaellum

Each of the five *Hbt. salinarum* archaellins, which together share almost 80% sequence identity, harbors three N-glycosylation sites, two of which are conserved and the third being shifted by two positions in ArlA2 (Fig. 2a b; Supplementary Fig. 2)[28,43]. The conserved N80 and N93 residues (following the numbering of ArlB1, hereafter) presented an extra density that forms dead-end protrusions extending from the filament surface (Fig. 2b). Given earlier mass spectrometry-based efforts demonstrating the glycosylation of these archaellin asparagine residues[28,29], our high-resolution structure thus directly confirmed that N80 and N93 are glycosylated. Indeed, we could fit three sugars of the N-linked tetrasaccharide (the composition of which was previously defined using NMR and MS)[26,27] known to decorate *Hbt. salinarum* archaellins at the N80 glycosylation site and two to three sugars at the N93 site (depending on the subunit), with no density being observed for the fourth sugar at either position, probably due to the flexibility of the glycan moiety, which projects beyond the protein surface of the filament. Differences in density dimensions at these two N-glycosylation sites likely stem from variations in the chemical environment created by neighboring residues, which affects the flexibility of each N-linked tetrasaccharide. Accordingly, N80 is found at the center of a β-sheet, while

N93 is part of a loop (Fig. 2a), such that the former is expected to be more rigid and produce a stronger signal in the cryo-EM map than the latter. The position of a third glycosylation site—N123—is conserved among four of the archaellins, with the position of this asparagine being shifted by two residues in ArlA2 (Fig. 2b; Supplementary Fig. 2). Density extending from N123, detected in only some archaellins, was much less pronounced than that seen next to N80 or N93, and as such, was unable to accommodate an entire glycan. Similarly, only a small density at the position of the comparable asparagine in ArlA2 was observed in some of the subunits (Supplementary Fig. 8). Therefore, we did not explicitly model the third tetrasaccharide linked to N123 (or the comparable asparagine in ArlA2). The presence of residual density protruding from N123, and in only some of the subunits, suggests that this region of the protein is flexible and differs among archaellins comprising the archaellum (Supplementary Figs. 2–4). As such, the signal from the N123-linked tetrasaccharide and its vicinity was lost upon averaging in reconstructing the cryo-EM map.

As expected, the N-linked tetrasaccharides are exposed to the solution (Fig. 2c). N93 is located near the interface of neighboring archaellins, raising the possibility that the bound glycan could participate in forming connections between archaellins. However, the

density extending from N93 protrudes from the protein surface, with no interactions between the attached tetrasaccharide and amino acids of adjacent archaellins being seen. N80 and N123 are both located away from interfaces with nearby archaellins, also making it unlikely that their linked glycans stabilize inter-molecular interactions. In the case of N80, this was confirmed by the density of the tetrasaccharide being distant from any inter-archaellin interface (Fig. 2c). In summary, given their distances and how they point away from inter-archaellin surfaces, the N-linked glycans decorating archaellins that comprise the *Hbt. salinarum* archaellum do not seem to play a significant role in promoting or supporting interactions between archaellins within an archaellum filament.

## Archaellum filament structures from the parent and N-glycosylation mutant strains are highly similar

To investigate the importance of archaellin N-glycosylation on archaellum filament structure, we determined high-resolution structures of archaellum filaments from two mutants that assemble truncated N-linked glycans. Specifically, cells deleted of *agl27* lack the fourth N-linked tetrasaccharide sugar (a sulfated glucuronic acid), while cells deleted of *agl26* also lack the third sugar (a sulfated iduronic acid)[27]. Archaella were isolated from the spent media of mutant strains as when isolating archaella from parent strain cells (see above), although larger volumes of spent medium were needed to obtain comparable amounts of material from the mutant strain cultures, in agreement with earlier SDS–PAGE experiments, which showed decreased amounts of archaellins in the mutant strain culture medium, as compared to a parent strain culture[29]. Cryo-EM data collection and structure determination, performed following the same workflow as used to determine the structure of wild-type archaellum filaments, yielded cryo-EM maps at average resolutions of 3.3 and 3.1 Å for Δ*agl27* and Δ*agl26* strain-derived archaellum filaments, respectively (Fig. 3; Supplementary Fig. 9).

Overall, the cryo-EM maps of the mutant strain archaellum filaments were reminiscent of the map of the wild-type archaellum filament. Even as early as the helical reconstruction step, the refined values of the helical rise and twist were very close to those obtained upon the same processing of wild-type filaments, suggesting that the organization of archaellins within an archaellum filament was not substantially altered by the lack of the terminal or the last two N-linked tetrasaccharide sugars (Fig. 3a; Supplementary Fig. 9). Indeed, super-positioning of the cryo-EM maps constructed without any applied symmetry revealed the three maps to fit well into each other (Fig. 3b).

As with the map of wild-type archaellum filaments, we focused our model building of filaments from each mutant strain on the central regions of the maps, which had the best quality (Supplementary Fig. 9c), yielding models of 25 subunits for archaellum filaments from both the Δ*agl27* and Δ*agl26* strains. Next, we compared the arrangement of archaellins within the filaments by aligning the polypeptide chains of the three models, without considering variations in N-linked tetrasaccharide conformation. Such alignment revealed the three models as matching almost perfectly, with very low root mean square deviation (RMSD) values of 0.62–0.78 Å (Fig. 3c). Consistent with the observation that the tetrasaccharides N-linked to archaellins that constitute wild-type archaellum filaments do not participate in inter-archaellin interactions (Fig. 2c), the absence of the fourth and/or third sugars of the tetrasaccharide did not affect the native packing of archaellins within the filament. We next specifically considered those regions surrounding the asparagine residues modified by glycosylation in a search for distinct local structural differences among archaellum filaments from the three strains. Barely any differences in the polypeptide chain in the three regions of N-glycosylation were observed across archaellum filaments from the parent and mutant strains. Any minor local shifts in backbone position and/or side-chain

conformation observed could be attributed to differences in the quality and/or resolution of the different cryo-EM maps. Therefore, our data cannot attest to any distinct structural changes that occurred within archaellins or their organization into archaellum filaments as a result of compromised N-glycosylation.

## Archaellins in N-glycosylation mutants display reduced densities adjacent to glycosylated asparagine residues

Whereas shortening of the N-linked glycans in the Δ*agl27* and Δ*agl26* strains yielded archaellum filaments exhibiting the same protein architecture as those in the parent strain, where the complete N-linked tetrasaccharide is assembled, the densities next to the decorated asparagine residues were reduced in the structures from the mutants (Fig. 4). Specifically, the density next to N80, which could accommodate three sugars in the map of a wild-type archaellum filament, had become smaller in the map of the Δ*agl27* strain filament, being now able to only fit two–three sugars. The same density was further reduced in the map of the Δ*agl26* strain archaellum filament, now accommodating only one sugar. The density nearby N93 fit two–three sugars in the map from the parent strain, yet only two sugars in the Δ*agl27* strain map. In the map from the Δ*agl26* strain, the density was further reduced and accommodated only one sugar. At the third glycosylation site (i.e., N123 in four of the five archaellins), we observed some density in the parent strain map, but less density in maps from the Δ*agl27* strain and barely any density in the map from the Δ*agl26* strain, such that not even a single glycan could be accommodated by the density near this position in either of the mutant strain maps. The decreased densities next to the glycosylated asparagine residues in our cryo-EM structures of mutant strain-derived archaellum filaments thus validate the roles of Agl26 and Agl27 in adding the third and fourth sugars, respectively, as previously elucidated using mass spectrometry[29].

## N-glycosylation mutants display jerky movements rather than unidirectional motility

Although truncation of the N-linked glycans decorating archaellum filaments had no observable effect on filament architecture, we, nonetheless, considered the effect of compromised N-glycosylation on swimming motility at the single-cell level, given previous reports of Δ*agl27* and Δ*agl26* strain cultures showing reduced motility in plate motility assays, relative to parent strain cultures, and failing to swim to the air–liquid medium interface, in contrast to parent strain cell cultures[29]. Accordingly, we imaged individual swimming cells from each strain using bright-field microscopy and tracked their swimming paths. Cells from the parent strain usually swam for several micrometers in an almost linear path at an average speed of $2.9 \pm 0.7 \, \mu m/s$ (Fig. 5; Supplementary Table 3; Supplementary Movie 1), consistent with the average swimming speed measured in an earlier study $(2.9 \pm 0.5 \, \mu m/s)$[50]. The average swimming speeds of cells from the two mutant strains were similar to the average swimming speed of parent strain cells (Supplementary Table 3). However, Δ*agl27* strain cells exhibited swimming in both single and ever-changing directions, while cells from the Δ*agl26* strain tended to swim in a restricted area and exhibit jerky movements (Fig. 5a). To more quantitatively describe the efficiency of swimming in reaching points distant from a starting point, we calculated the average confinement ratio, defined as the ratio between displacement (i.e., the straight-line distance between the starting and end points of the swimming track) and path length (i.e., the total distance traveled) for each strain, with more unidirectional and efficient motility yielding a higher ratio. On average, this measure was some two-fold lower for Δ*agl27* and Δ*agl26* strain cells ($0.14 \pm 0.01$ and $0.17 \pm 0.01$, respectively) than for parent strain cells ($0.30 \pm 0.02$), reflecting the localized motility of the former and the more consistent swimming paths of the latter (Fig. 5b; Supplementary Table 3). Similarly, the average maximum distance traveled (i.e., the distance

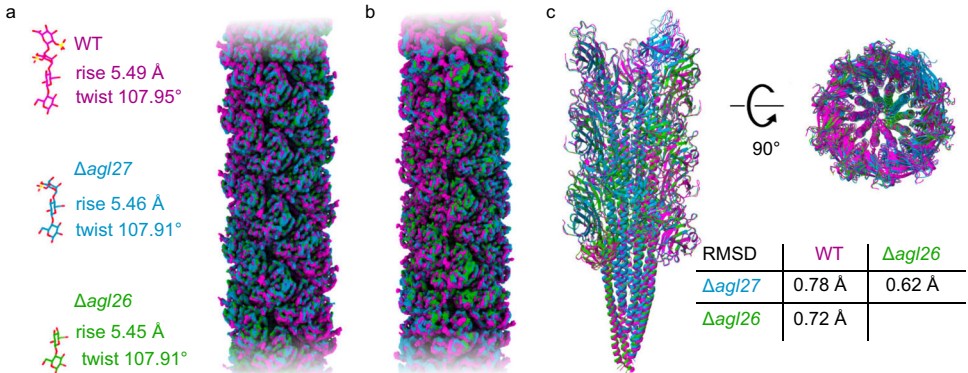

**Fig. 3 | Archaellum filament structures from the parent strain and N-glycosylation mutants are highly similar. a** Super-positioning of helical reconstructions of archaellum filaments from the parent (WT) (magenta), Δagl27 (blue), and Δagl26 (green) strains, showing their overlap. The refined helical rise and twist values are very close to each other, as indicated for each cryo-EM map. The figure focuses on the central regions of the cryo-EM maps, which exhibited the best quality (Supplementary Figs. 1e and 9c). **b** Super-positioning of the symmetry-free cryo-EM maps obtained for archaellum filaments from the three strains. See Supplementary Fig. 9 for more details. **c** Alignment of the three models built into the maps in (**b**). Root mean square deviation (RMSD) values for each pair of models indicated in the table at the bottom right were calculated using "super" in PyMol (http://www.pymol.org). N-linked glycans were not considered in the RMSD calculations.

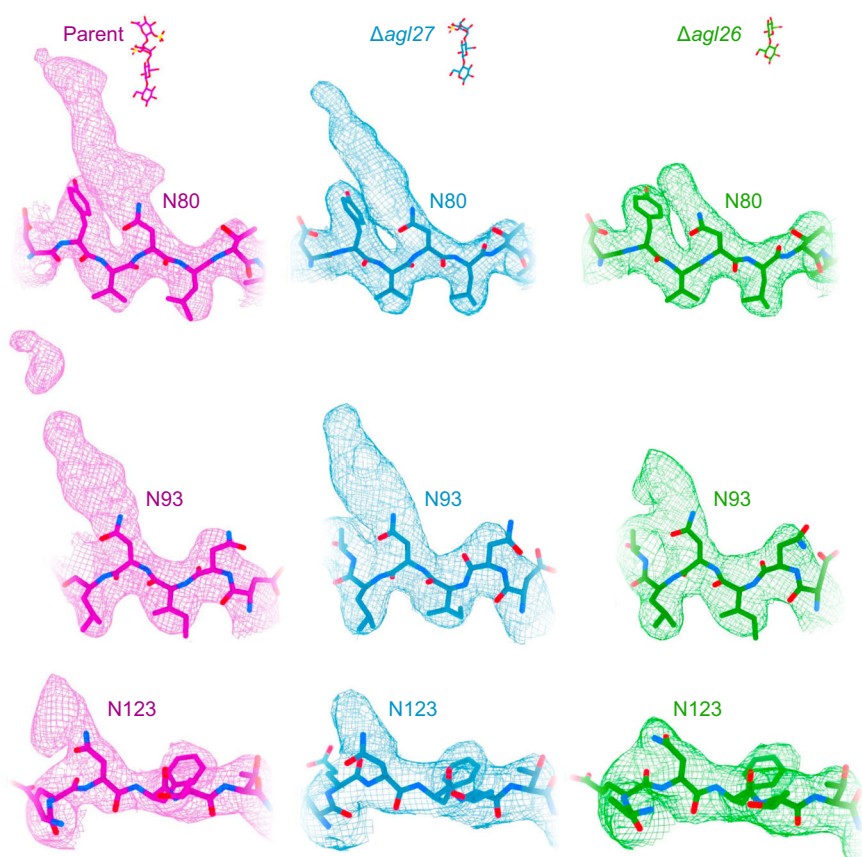

**Fig. 4 | Archaellins in N-glycosylation mutants display reduced densities adjacent to modified asparagine residues.** Cryo-EM maps (shown as meshes) and models of the close vicinities of the three N-glycosylation sites in *Hbt. salinarum* archaellins from the parent (WT) (magenta), Δagl27 (blue), and Δagl26 (green) strains. Glycans modeled at positions N80 and N93 were removed to allow a clear view of the density reductions adjacent to these residues in archaellins from the N-glycosylation mutant strains.

between the starting point of the track and the furthest point reached), was almost two-fold higher for parent strain cells ($5.9 \pm 0.4\,\mu m$) than for Δagl27 and Δagl26 strain cells ($3.0 \pm 0.1\,\mu m$ and $3.4 \pm 0.3\,\mu m$, respectively).

Finally, the parent and mutant strain cells differed not only in terms of their swimming patterns but also in terms of their interactions with other cells. While cells from the parent strain generally swam individually, with pairs of cells swimming together only being observed occasionally, clumps containing several cells displaying vibrational movements were often seen in samples of the N-glycosylation mutant strains (Supplementary Movie 1). These immotile cell clumps may have contributed to the overall deterioration

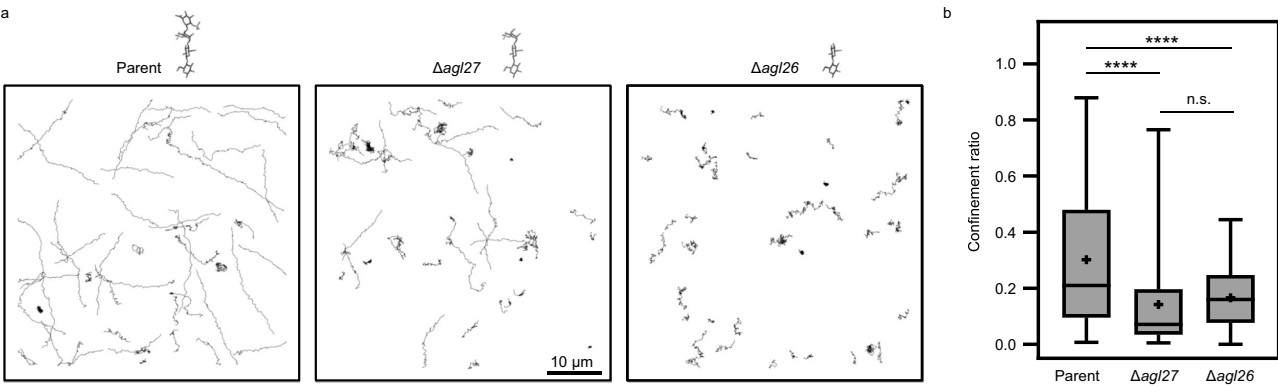

**Fig. 5 | N-glycosylation mutants display jerky movements rather than uni-directional motility. a** Representative swimming paths of cells from the parent, Δ*agl27*, and Δ*agl26* strains, taken from Supplementary Movie 1. **b** Box-and-Whisker plot of confinement ratios (displacement/total distance traveled) measured for parent ($n = 133$), Δ*agl27* ($n = 163$), and Δ*agl26* ($n = 65$) strain cells.

Whiskers represent the first and last quartiles and extend to the maximum and minimum points. Vertical lines indicate medians and plus signs indicate averages. Statistical significance was determined using a one-way analysis of variance with a Holm–Šídák test. ****$P < 0.0001$, n.s. non-significant. Source data are provided as a Source Data file.

in motility observed in earlier assays assessing populations of the N-glycosylation mutant strains[28–30].

## Shortening the N-linked tetrasaccharide results in archaellum filament clustering

An explanation for why the structure of archaellum filaments from the parent and N-glycosylation mutant strains presented no differences, yet motility was clearly compromised as a result of N-linked tetra-saccharide truncation, was found in cryo-EM micrographs, where a noticeable difference distinguishing archaellum filament packing in the parent and mutant *Hbt. salinarum* strains was observed. After differentially diluting aliquots of spent growth medium from the parent, Δ*agl27*, and Δ*agl26* strains to account for differences in the degree of archaellum filament release in each case[29], cryo-EM micrographs revealed that whereas wild-type archaellum filaments were well-separated and usually straight, archaellum filaments from the Δ*agl27* and Δ*agl26* strains tended to bundle and curve (Fig. 6a; Supplementary Fig. 10a). Further diluting the Δ*agl27* and Δ*agl26* strain archaella-containing samples did not lessen such bundling, indicating archaellum filament clustering to be an intrinsic property of these preparations and not due to filament concentration. Moreover, despite the fact that all three samples contained the same salt concentration, archaellum filament bundling was not observed in the parent strain-derived sample, thereby arguing against such clustering being an artifact related to the level of sample salinity.

To further validate that archaellum filament clustering was not an artifact of the isolation procedure used, we imaged intact parent and Δ*agl27* strain cells grown and maintained in 4 M NaCl using negative-stain EM (Fig. 6b c). We categorized those cells that harbored more than one archaellum filament into two groups, namely, cells presenting archaellum filaments that were physically separated from each other and cells presenting two or more archaellum filaments twisted around one another. In the case of parent strain cells ($N = 64$), 35% of the cells exhibited two or more physically separated archaellum filaments, with only 6% of the cells exhibiting filaments twisted together. In contrast, almost half of the Δ*agl27* cells examined ($N$=84) presented more than two archaellum filaments twisted around each other (49%), with a smaller population of cells presenting separated multiple archaellum filaments (25%). Hence, the bundling of archaellum filaments, whether isolated or still attached to the cell, occurred far more frequently as a result of compromised N-glycosylation. At the same time, 42% of parent strain cells presented only a single archaellum filament, as opposed to only 19% of the Δ*agl27* cells. Finally, several parent cells did

not present any archaella (17%), as these entities are presumably prone to detachment from the cell during growth or shearing during sample preparation. Δ*agl27* cells were more resilient to this phenomenon, with only 7% of the cells lacking archaella. This observation is consistent with the need to grow larger volumes of mutant strain cells, as compared to parent strain cells, so as to obtain comparable amounts of isolated archaella from the spent medium[29].

Finally, to gain insight into the nature of archaellum filament bundling by addressing the interactions formed between filaments within a bundle, we re-picked particles from the cryo-EM micrographs of the N-glycosylation mutant strains to include two filaments per particle. Averaging these two filament-containing images resolved details in one filament of the pair as obtained when averaging single filaments (Supplementary Figs. 1c and 9a), whereas the second filament in the pair appeared blurred (Supplementary Fig. 10b). If filaments within bundles were to have formed distinct and specific interactions through specific filament regions, we would have expected the averages to coincide into an image that resolved details in both filaments in the pair. Instead, it seems that aligning particles with respect to one filament of the pair resulted in the second filament being detected at a different position in different particles, thereby creating an overall obscured image of that filament. These data suggest that the clusters formed from archaellum filaments devoid of the third and/or fourth sugars of the N-linked tetrasaccharide are not stabilized by distinct interactions, but rather exhibit random "sticking" of protein interfaces.

In summary, the absence of the last or of the last two sugars of the N-linked tetrasaccharide decorating *Hbt. salinarum* archaellins results in archaellum filament clustering that compromises swimming motility.

## Discussion

Protein N-glycosylation is an almost universal post-translational modification that affects cell surface proteins in Archaea. In addition to the S-layer glycoproteins that comprise the cell wall of many archaeal species, various filamentous structures that protrude from the cell, such as pili used for adhesion and/or twitching motility and archaella that drive swimming motility, are also assembled from N-glycosylated proteins, i.e., pillins and archaellins, respectively[2,44,51–56]. The truncation or absence of N-linked glycans that decorate archaellins often results in compromised swimming[28–35]. Thus, the conclusion from some of these studies was that the N-linked glycans on the surface of archaellins (or some of the sugars comprising these structures) promote

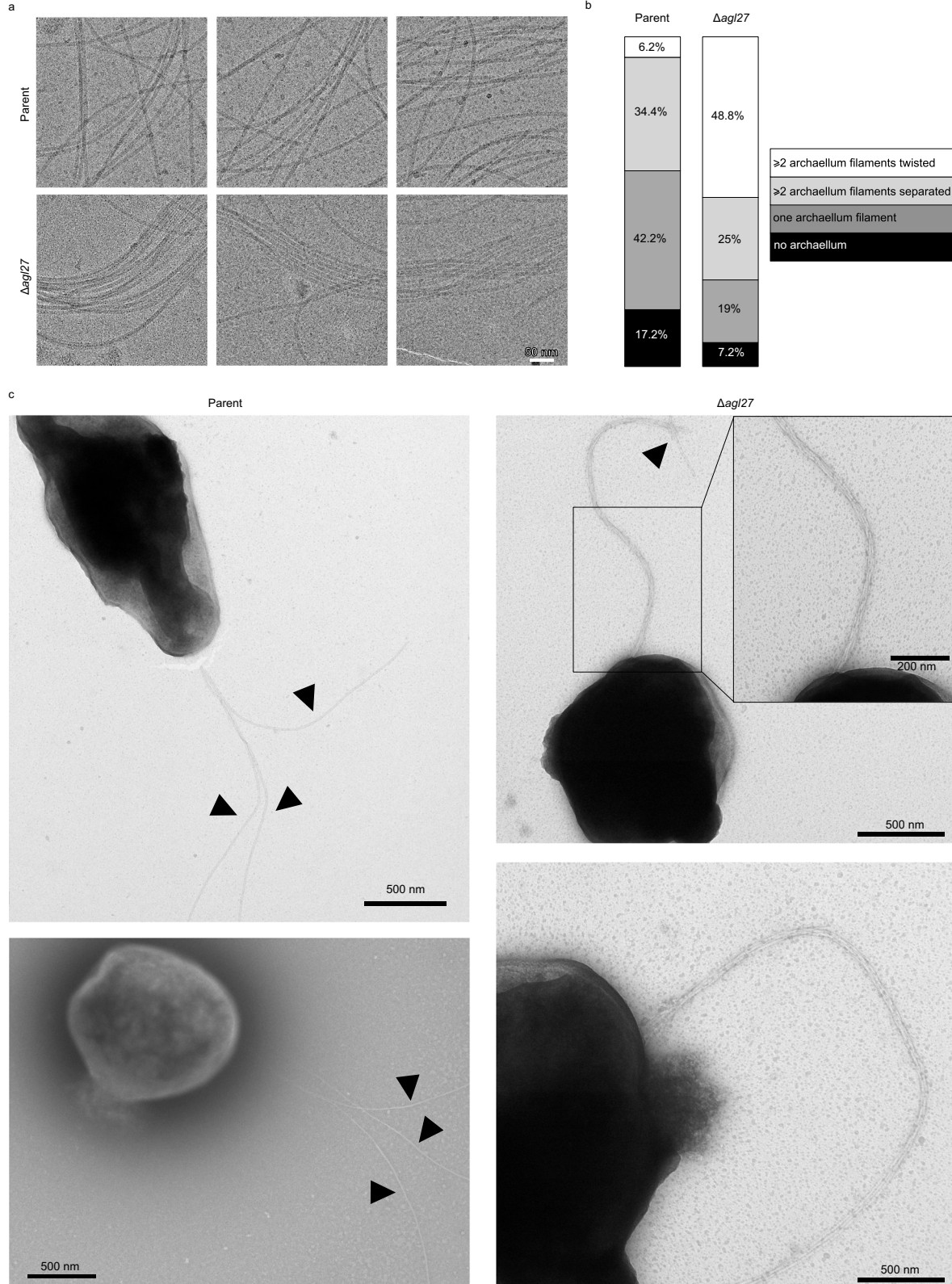

archaellum filament assembly and stability, which in turn, facilitates motility. In the present work, *Hbt. salinarum* parent and N-glycosylation mutant strains were used to test these concepts by directly examining the contributions of N-glycosylation to archaellum filament structure and function.

The structure of the *Hbt. salinarum* archaellum filament presented here (Fig. 1) exhibits structural features similar to those seen in archaellum filaments from methanogenic, hyperthermophilic, and acidophilic archaea[37–40,42]. These earlier structures reported putative sites of archaellin glycosylation, identifying dead-end protrusions next to asparagine (i.e., N-glycosylation), and serine and threonine (i.e., O-glycosylation) residues, with the number of glycosylation sites ranging between three and eight sites per archaellin. In the case of *Hbt. salinarum* archaellins, those asparagines modified by a tetrasaccharide of

**Fig. 6 | Shortening of the N-linked tetrasaccharide results in archaellum filament clustering. a** Representative cryo-EM micrographs of archaellum filament samples from the parent (top panels) and Δ*agl27* (bottom panels) strains. Comparable micrographs of archaellum filaments from Δ*agl26* strain cells are presented in Supplementary Fig. 10a. For each sample, micrographs were recorded from five grids, prepared from three independent samples. **b** Distribution of parent (n = 64) and Δ*agl27* strain cells (n = 84) according to four categories: no archaellum, one archaellum filament, separated multiple archaellum filaments, and bundled multiple archaellum filaments. Source data are provided as a Source Data file.

**c** Representative negative-stain EM micrographs of parent and Δ*agl27* strain cells with more than one archaellum filament. Left – in most of the parent strain cells that presented two or more archaellum filaments, the filaments were separated from each other, as indicated by arrowheads. Right – almost half of the Δ*agl27* strain cells presented twisted and bundled archaellum filaments. Here, the arrowhead indicates a point at which at least three archaellum filaments unwound from the bundle. Inset – zoom into the region marked with a rectangle. The experiment was repeated independently three times for the parent strain and four times for the Δ*agl27* strain.

defined content and architecture[27] were previously identified[28,29], such that our high-resolution structure validated these earlier findings. Furthermore, the structure revealed that the N-linked tetrasaccharides point outwards, protruding the protein surface, and hence that the glycan terminal sugars, which were not observed in most subunits due to averaging of their positions in the cryo-EM map, are quite flexible (Fig. 2). Indeed, the fact that not all of the tetrasaccharide sugars could be resolved in the cryo-EM map argues that the N-linked tetrasaccharides do not form specific stable interactions with adjacent archaellins that would contribute to archaellum filament assembly.

The availability of *Hbt. salinarum* N-glycosylation pathway mutants enabled us to conduct the first high-resolution study assessing how perturbing this post-translational modification affects archaellum filament assembly and structure. In possible agreement with earlier structure-based predictions that N-glycosylation is required to stabilize interactions between archaellins[41], we observed that at least one glycosylated *Hbt. salinarum* archaellin asparagine residue (i.e., N93) is located near an inter-archaellin interface (Fig. 2). Yet, we did not observe any significant changes in the structure of archaellins or their packing when the N-linked tetrasaccharides were replaced by their tri- or disaccharide precursors in the Δ*agl27* and Δ*agl26* strains, respectively (Fig. 3), indicating that archaellin folding and polymerization do not depend on the two terminal tetrasaccharide sugars. At the same time, we cannot rule out that the first and second sugars are somehow required for archaellin folding, stabilization, and/or packing. Indeed, the deletion of oligosaccharyltransferase-encoding *aglB* not only abolished archaellin N-glycosylation but also resulted in a complete loss of archaella in *Hbt. salinarum*[28], as well as in *Methanococcus voltae*[31], *Methanococcus maripaludis* (*M. maripaludis*)[57], and *Haloferax volcanii* (*Hfx. volcanii*)[34]. Likewise, the deletion of genes encoding glycosyltransferases that add the first and/or second N-linked glycan sugars in these species also prevented archaellum assembly[29,30,32,34,57]. Moreover, mutating all four archaellin asparagine residues destined for glycosylation prevented archaellum assembly in *M. maripaludis*[58]. In conclusion, while the requirements of N-glycosylation for archaellum filament assembly and stability are not entirely understood, it is clear that the two terminal sugars of the N-linked tetrasaccharide decorating *Hbt. salinarum* archaellins are not essential for archaellation or archaellum filament assembly.

Since the absence of the third and fourth sugars of the N-linked tetrasaccharide decorating *Hbt. salinarum* archaellins did not affect archaellum filament structure yet did reduce motility (Fig. 5)[29], it would seem that a property other than archaellum filament assembly is adversely affected in Δ*agl27* and Δ*agl26* strain cells. Archaellum filaments decorated with truncated N-linked glycans in these strains bundled and twisted around each other, whether isolated or attached to cells, unlike wild-type filaments, which were usually well-separated (Fig. 6). Cells from the parent strain, moreover, presented no or only a single archaellum filament more frequently than did Δ*agl27* strain cells, which, conversely, presented filament bundling to a far greater extent than seen with the parent strain. Furthermore, the appearance of vibrating cell clumps in Δ*agl27* and Δ*agl26* strain cultures, but not in parent strain cultures, may be a result of bundling of archaellum

filaments from neighboring cells (Supplementary Movie 1). It is unlikely that the enhanced filament bundling seen with the mutant strains is due to increased archaellin synthesis, relative to the parent strain, given our earlier qRT-PCR results, which showed that the transcription of *Hbt. salinarum* archaellins was either unaltered or reduced in Δ*agl27* and Δ*agl26* cells[29]. However, we cannot rule out completely that bundling in the N-glycosylation mutant strains is also affected by changes in the relative expression of the various archaellins. Finally, bundling may be a result of non-specific hydrophobic sticking, rather than defined and specific interactions between filaments (Supplementary Fig. 10b). Indeed, the surfaces of ArlA2, ArlB1, and ArlB3 contain several hydrophobic patches (Supplementary Fig. 7). Still, detailed understanding of how filament bundling occurs will require further investigation.

The link between the observed archaellum filament bundling and reduced motility could be explained by a study of *Hbt. salinarum* swimming conducted at the single-cell level[50]. There, it was noted how archaellum filaments repeatedly bundled and separated, as well as rotated independently in opposite directions, during swimming. The tendency of archaellum filaments from the Δ*agl27* and Δ*agl26* strains to cluster would presumably prevent the filament separation needed for effective swimming, thereby hindering unidirectional cell motility (Fig. 5). In contrast, in the parent strain, where archaellins are modified by the complete N-linked tetrasaccharide, archaellum filaments would be less prone to such bundling, suggesting that the N-linked tetrasaccharides protruding from the archaellum filament surface play an active role in preventing filament aggregation. Since archaellum filament bundling was the most pronounced phenotypic change resulting from the perturbation of the N-glycosylation pathway addressed in the present study, we propose such bundling to be the direct cause of the disrupted motility seen with the Δ*agl27* and Δ*agl26* strains (Fig. 5)[29]. Future single-cell analyses of archaellum filaments during swimming[50] will be useful to establish a direct connection between filament bundling and perturbed motility in these mutant strains. The proposed function of protein-bound glycans in preventing filament clustering could be facilitated by either or both of the two distinct properties of the *Hbt. salinarum* N-linked tetrasaccharide. First, the negative charges of the iduronic acid at position three and the glucuronic acid at position four of the N-linked tetrasaccharide decorating archaellins are augmented by the sulfation of each of these sugars[27]. Such concentrated negative charge could drive separation of the archaellum filaments through electrostatic repulsion. While it is true that the protein surface of the archellum filament presents large regions of negative potential (Supplementary Fig. 7), such distribution of negative charges does not seem to be sufficient to prevent filament bundling. Second, the length of the N-linked tetrasaccharides, which allows them to protrude the archaellum filament surface, would enable them to act as physical spacers that physically distance and thus prevent contacts between neighboring filament surfaces that would lead to bundling mediated by protein–protein interactions (Fig. 7). The impact of N-glycosylation on maintaining archaellum filament separation via electrostatic repulsion and/or steric hindrance might explain why the introduction of additional N-glycosylation sites into *M. maripaludis* archaellins resulted in a hyper-motile strain[58].

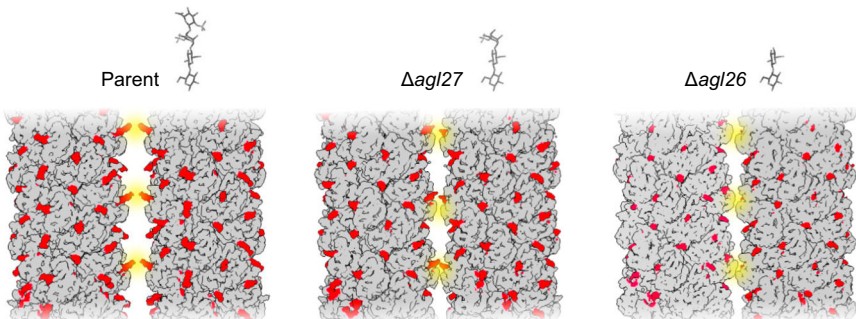

**Fig. 7 | A schematic model describing the impact of N-linked tetrasaccharide trimming on interactions between archaellum filaments.** Cryo-EM maps of archaellum filaments from the three strains addressed here are shown, with the densities corresponding to N-linked glycans colored red. Two maps of filaments from each strain were manually placed side by side to illustrate their possible interactions and potential clashes (depicted as yellow halos). While the N-linked tetrasaccharides associated with wild-type archaellum filaments would be the first point of contact between filaments and thus prevent protein–protein interactions due to electrostatic repulsion and/or steric interference, the shorter tri- or disaccharides attached to filaments from the Δagl27 and Δagl26 strains, respectively, would not be able to prevent such interactions to the same extent. As a result, wild-type archaellum filaments can separate, whereas filaments comprising archaellins bearing truncated N-linked glycans in the mutant strains would instead bundle.

A similar phenomenon of filament clustering was observed in *Hfx. volcanii*, where disruption of pilin glycosylation via *aglB* deletion led to pili aggregation and microcolony formation[59]. In the case of adhesion pili, maintaining pili separation from each other is necessary for surface adhesion, rather than the microcolony formation that results from the clustering of pili from neighboring cells. Observations of filament clustering not only in different archaea but also of different types of archaeal cell surface filaments suggest that the minimization or even prevention of filament bundling is a universal role of the N-linked glycans decorating these structures.

In conclusion, addressing the link between N-glycosylation and cell motility in this study provided valuable insight into the role of the N-linked tetrasaccharide decorating *Hbt. salinarum* archaellins in promoting swimming, specifically, the importance of the third and fourth tetrasaccharide sugars for such activity, thus offering direct evidence for how N-glycosylation affects *Hbt. salinarum* physiology. Still, numerous questions remain unanswered. Why does the absence of N-glycosylation or the addition of only a single N-linked sugar prevent archaellin filamentation? Why does *Hbt. salinarum* express five different archaellins? Can *Hbt. salinarum* modulate the archaellin content of the archaellum under certain conditions, and if so, what are the relative advantages of different archaellin combinations? Does *Hbt. salinarum* control the extent of N-glycosylation and/or N-linked glycan length in a physiologically relevant manner? As in this study, high-resolution structural biology of archaella derived from mutant strains perturbed in archaellum filament content, assembly, and/or motility, together with analyses of cell swimming patterns, will be valuable in answering these open questions.

## Methods

### Cell growth

*Hbt. salinarum* parent and mutant strain[28,29] cells were grown in a medium containing 250 g NaCl, 20 g MgSO$_4$·7H$_2$O, 3 g sodium citrate, 2 g KCl, and 10 g peptone/l, supplemented with 50 μg/ml uracil, at 42 °C[60]. The cultures were grown for 3 days until an OD$_{600}$ of 1.3 was reached.

### Enrichment of archaella

*Hbt. salinarum* archaella were enriched from spent growth medium as previously described[28]. Briefly, cultures were grown to logarithmic (OD$_{600}$ ~ 0.8) phase and held at room temperature without shaking for 24 h. The cultures were centrifuged for 30 min (6000 × g, 15 °C). The supernatant (post-spin 1 supernatant) was collected and centrifuged again for 15 min (16,000 × g, 15 °C). The supernatant (post-spin 2 supernatant) was centrifuged for 2 h (40,000 × g, 4 °C). The pelleted material (post-spin 3) was resuspended by shaking in 1 ml of 4 M basal salt solution (250 g NaCl, 20 g MgSO$_4$·7H$_2$O, 3 g sodium citrate, 2 g KCl/l) and heated for 10 min at 90 °C. The heated suspension was centrifuged for 15 min (16,000 × g, 15 °C). The resulting supernatant (post-spin 4 supernatant) was maintained at 4 °C for 24 h and then centrifuged for 2 h (40,000 × g, 4 °C). After removal of the supernatant, the pellet (post-spin 5 pellet) was resuspended in 60 μl of a solution containing basal salt solution and water at a 1:4 ratio.

### Sample preparation for cryo-EM

Archaella were derived from a parent strain culture (200 ml), and from Δagl27 and Δagl26 strain cultures (400 ml each). Re-suspended solutions (as described above) of parent, Δagl27, and Δagl26 strain-derived archaella samples were diluted 20-fold, 8-fold, and 2-fold, respectively, before application onto grids. Three microliter aliquots of each sample were deposited on glow-discharged Quantifoil R 1.2/1.3 holey carbon grids (Quantifoil, Großlöbichau, Germany). The sample-bearing grids were manually blotted for 4 s at room temperature and vitrified by rapid plunging into liquid ethane using a home-built plunging apparatus. The frozen samples were stored in liquid nitrogen until imaging.

### Cryo-EM data acquisition

Cryo-EM datasets were collected under cryogenic conditions on a Glacios microscope operated at 200 kV and equipped with a Falcon 4i Direct Electron Detector coupled to a Selectris X energy filter (Thermo Fisher Scientific) set at ±5 eV from the zero-loss peak. Movies were recorded in a dose-fractionated counting mode using EPU (Thermo Fisher Scientific) with a pixel size of 0.89 Å and a total electron dose of 40 e$^-$/Å$^2$. Data were collected at a de-focus range of −0.5 to −1.5 μm. Further data collection statistics are reported in Supplementary Table 1.

### Cryo-EM data processing

Dose-fractionated image stacks were imported into cryoSPARC (v4.3.0)[61] and processed according to the workflow summarized in Supplementary Figs. 1 and 9. Image stacks were subjected to patch-based motion correction and patch-based contrast transfer function (CTF) estimation. For further processing, micrographs with a better than 4.5-Å CTF fit resolution were chosen, resulting in 1296 micrographs retained from an initial 5054 of parent strain archaellum filaments, 3714 micrographs from 4846 of Δagl27 strain filaments, and 4616 micrographs from 5512 of Δagl26 strain filaments. Particles were picked using the filament tracer with a separation distance of 45 Å

between boxes, which were extracted at a box size of 370 Å. Initially, 455,539, 1,045,640, and 1,333,166 overlapping particles of parent, Δ*agl27*, and Δ*agl26* strain-derived archaellum filaments, respectively, were obtained. Following two-dimensional classification, 334,202, 265,906, and 448,635 particles of parent, Δ*agl27*, and Δ*agl26* strain-derived archaellum filaments, respectively, were chosen for helical refinement. Based on previous negative-stain EM studies on archaellum filaments from *Hbt. salinarum*[36], the initial parameters applied in helical refinement were a twist of 108° and a helical rise of 5.4 Å, which yielded maps with average resolutions of 3.4, 3.6, and 2.9 Å for parent, Δ*agl27*, and Δ*agl26* strain-derived archaellum filaments, respectively. These helical reconstructions were low-pass filtered to 30 Å resolution and used as input volumes in non-uniform refinement of the same sets of particles. In this last round of refinement, no symmetry was applied, and de-focus refinement and global refinement were performed. Overall resolutions of the maps (3.2 Å for parent strain, 3.3 Å for Δ*agl27* strain, and 3.1 Å for Δ*agl26* strain archaellum filaments) were estimated using the gold-standard Fourier shell correlation criterion (FSC = 0.143). Maps were sharpened using B-factor in cryoSPARC. For visualization purposes alone (i.e., for interpretation of the maps during model building and preparation of figures), maps were sharpened using DeepEMhancer[62] or EMReady[63]. Statistical information for the final density maps is presented in Supplementary Table 1.

## Model building

Initial models for the five *Hbt. salinarum* archaellins were generated using AlphaFold2[64] and were docked into the cryo-EM map of the parent strain archaellum filament using UCSF ChimeraX[65]. After thorough examination of various regions of the map, we could not resolve which specific archaellins comprise the archaellum filament and built a model of 26 identical subunits in COOT[66]. The consensus subunit model followed ArlB1 residue numbering and included residues 13–193.

Positions at which the five archaellins share the same sequence were explicitly modeled, whereas positions that differ among the archaellins were usually modeled as alanine residues (UNK). When large side-chain density was observed and could not be accounted for only by an alanine, the corresponding ArlB1 residue was modeled. Supplementary Table 2 lists the positions that differ among archaellins and describes how they were modeled in the consensus model.

A crystallographic information file was generated for the N-linked tetrasaccharide in AceDRG[67] within CCP4 based on NMR studies of the *Hbt. salinarum* N-linked tetrasaccharide[27]. The tetrasaccharide model was linked to the relevant asparagine residues (N80 and N93 according to ArlB1 residue numbering) in COOT using AceDRG. For N123, which is shifted by two residues in ArlA2, density was not always observed and, therefore, the tetrasaccharide was not modeled.

To build models of archaellum filaments from the Δ*agl27* and Δ*agl26* strains, the parent strain model was used as an initial model and the subunits were repositioned and rebuilt according to the maps in COOT. The terminal sugar was removed from the N-linked tetrasaccharide, retaining a trisaccharide in the model of Δ*agl27* strain-derived archaellum filament, and an additional sugar was removed in the model of Δ*agl26* strain-derived archaellum filament, which retained a disaccharide.

All models were real-space refined against the corresponding maps (sharpened by B-factor, and not AI-based methods) using PHENIX[68], iteratively rebuilt in COOT, and refined in PHENIX until completion. Model validation was performed with MolProbity[69]. Molecular graphics figures were prepared using UCSF ChimeraX[65].

## Imaging and analysis of cell motility

Cells were grown to $OD_{600} \sim 0.8$ and diluted two-fold with a basal salt solution before application into the wells of a 96 round well μ-Plate (Ibidi, Grafelfing, Germany). Bright-field imaging of cells was performed on a 3i Marianas (Denver, CO) confocal microscope equipped with a Prime 95B sCMOS camera, a 100× Zeiss Plan-Apochromat oil, and a 1.4 NA objective. Movies of 350 frames were recorded at a frame rate of 12.5 Hz using SlideBook. For each sample, cells from at least two independent cultures, imaged in at least three independent wells per culture, were imaged. Cell segmentation was performed using CellPose[70] and tracking was performed using TrackMate[71] in ImageJ. Swimming speeds, confinement ratios, and maximum distances were calculated from tracks that were at least 50 frames long. Statistical analysis was performed in Prism 10.2.2 (GraphPad). One-way analysis of variance with a Holm–Šídák test was used to determine statistical significance.

## Negative-stain imaging of intact cells

Parent and Δ*agl27 Hbt. salinarum* strain cells were prepared as previously described[28]. Three microliter aliquots were applied to glow-discharged 300 mesh copper grids covered by a thin layer of continuous carbon type-B film (Ted Pella) and stained with 2% uranyl acetate. The grids were imaged on an FEI Talos F200C microscope (Thermo Fisher Scientific) operated at 200 kV at a nominal magnification of ×17,500 to ×73,000 using a Ceta 16 M pixel CMOS camera (Thermo Fisher Scientific). Images were recorded using Velox (Thermo Fisher Scientific). For each sample, cells from three individual cultures were imaged.

## Reporting summary

Further information on research design is available in the Nature Portfolio Reporting Summary linked to this article.

## Data availability

The atomic coordinates generated in this study for the *Hbt. salinarum* archaellum filament from the parent, Δ*agl27*, and Δ*agl26* strains have been deposited in the Protein Data Bank (https://www.rcsb.org/) under accession numbers 9EQ7, 9ETU, and 9ESM, respectively. The electron density maps for the archaellum filaments from parent, Δ*agl27*, and Δ*agl26 Hbt. salinarum* strains have been deposited in the EM DataResource (https://www.emdataresource.org/) under the accession numbers EMDB–19905, EMDB–19962, and EMDB–19943, respectively. Source data are provided with this paper.

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

## Acknowledgements

We thank Dr. A. Upcher for assistance with imaging negative-stained cells, Dr. U. Hadad for assistance with imaging cell motility, D. Sevilla Sanchez for assistance with motility analysis, and Y. Barouch for high-performance computing support. This research was supported by a grant from the Israel Science Foundation (414/20) to J.E.

## Author contributions

I.G.-H. and J.E. conceived the research, planned the experiments, and wrote the paper. Z.V., L.M., R.Z., and I.G.-H. collected the data. S.S., A.S., and I.G.-H. analyzed the motility and cryo-EM data. J.E. secured funding.

## Competing interests

The authors declare no competing interests.
