## [Peer Review File · Nature Communications]

Perturbed N-glycosylation of Halobacterium salinarum
archaellum filaments leads to filament bundling and
compromised cell motilityEditorial Note: This manuscript has been previously reviewed at another journal that is not operating a transparent peer review scheme. This document only contains reviewer comments and rebuttal letters for versions considered at *Nature Communications*.

Reviewer #1 (Remarks to the Author):

The revised manuscript with the updated title "Perturbed N-glycosylation of Halobacterium salinarum archaellum filaments leads to filament bundling and compromised cell motility" by Sofer et al. is a substantially improved version of the original manuscript, and in addition to the detailed structural analysis of Halobacterium salinarum archaella now presents data on the motility of individual cells. The manuscript focuses on the impact of impaired N-glycosylation on both archaella structure and resulting effects on cell motility. This revision includes additional details on the already commendable structural analysis of the archaellum filaments. In addition, the new motility analyses of individual cells provide valuable insights into the swimming behavior of cells with perturbed N-glycosylation. Unfortunately, despite the additional data, the functional insights are still very limited.

Major Comments:

1. In their response the authors stated that "earlier efforts attributed the loss of motility to an inability or compromised ability to assemble archaellum filaments by the different N glycosylation mutants.", and in the manuscript similar statements (e.g. "The explanation offered for these observations was that compromised N-glycosylation affects archaellum assembly and structure, and by extension, function, reflected as compromised swimming ability.", and "As such, the N-linked glycans on the surface of archaellins were assigned roles in promoting archaellum filament assembly and stability, which in turn, facilitates motility") are made. The authors then contrast these previous findings with their analysis of archaella structure and cell motility for strains lacking the last two sugars of the N-linked tetrasaccharide in Hbt. salinarum to highlight that archaellum filament assembly or structure was not significantly affected. However, this comparison is a misleading representation of previous findings. Previous studies not only in Hbt. salinarum, but also in Hfx. volcanii, Methanococcus voltae, and Methanococcus maripaludis indicated that motility is only fully lost in some N-glycosylation pathway mutants (e.g. if the OST AgIB is deleted, or enzymes involved in the attachment of the first monosaccharides) while other N-glycosylation pathway mutants (often for enzymes involved in the attachment of the last monosaccharide(s)) exhibited reduced motility, implying that a functional archaellum is still assembled. The manuscript presented here confirms these findings on the structural level, but the distinction between motile and non-motile N-glycosylation pathway mutants, and their corresponding implications on archaellum assembly need to be stated more clearly in both the introduction and discussion, to make clear that previous studies already showed that not all monosaccharides are needed for archaellum assembly, and that the results of the current study do not contradict previous findings about the disrupted archaellum assembly e.g. in agIB mutants.
2. The two main findings that are truly novel are the bundling of archaellum filaments, and the altered single cell swimming behavior. However for both of these findings, the mechanistic reasons remain to be elucidated, and potential explanations that would represent alternatives to the suggested electrostatic repulsion or spatial hindrance are inappropriately discussed (see subsequent comments). As such, the functional or mechanistic insights provided by this study are rather limited. The study reveals filament bundling, but cannot clarify why or how this bundling occurs. Furthermore, the study shows that previous findings on the reduced motility of N-glycosylation pathway mutants is due to altered swimming behavior, but without elucidating the movement of the corresponding (potentially bundled) archaella.
3. Results on the number of archaellum filaments in the different strains are presented in a confusing way. In their comments, the authors state that their Fig 6 "misled the reviewer into concluding that the mutant strains produced higher levels of archaellum filaments than did the parent strain". However, the current data still suggest said higher levels of archaellum filaments in the mutant strains. Based on the authors source data for Fig 6, a significant difference in the number of archaella per cell was observed between the WT (an average of 1.6 filaments per cell) and the agI27 mutant strain (an average of 2 filaments per cell). It is unclear why this increase in filaments per cell would not potentially contribute to the observed increase in bundling. The authors also state that the mutants show a decreased shearing of archaella during sample preparation (or during growth), but it is unclear why the perturbed N-glycosylation (and/or the bundling) would lead to an altered shearing behavior. Finally, the authors also state that "It is unlikely that the enhanced filament bundling seen with the mutant strains is due to increased archaellin synthesis, relative to the parent strain, given our earlier qRT-PCR results, which showed that the transcription of Hbt. salinarum archaellins was

either unaltered or reduced in Δ agl27 and Δ agl26 cells". However, these previous results showed unaltered transcription for 2-3 of the 5 tested archaellins, and the corresponding protein abundances have not been tested (e.g. via quantitative mass spectrometry). A difference in protein abundance therefore cannot be ruled out. Furthermore, since the identity of the individual subunits in the filament could not be revealed, it is possible that the ratio of different archaellins in the filament changed and thereby affected the observed curvature and bundling.

4. A direct connection between the altered swimming behavior and the bundling of filaments is implied but not confirmed through experimental evidence. It is unclear whether the observed swimming cells in the mutant have bundled archaella, or whether they are examples of cells with single/separated archaella. This distinction would be important to conclude whether the bundling itself is linked to the altered swimming behavior, or whether additional factors play a role.

Minor comments:

1. "Likewise, deletion of genes encoding the glycosyltransferases adding the first and second N-linked glycan sugars in *Hbt. salinarum*^{29,30} and *M. voltae*³² also prevented archaellum assembly." It should be noted that the same was also observed for *Methanococcus maripaludis* (VanDyke et al 2009) and *Haloferax volcanii* (Tripepi et al 2012)

2. "Moreover, mutating all four archaellin asparagine residues destined for glycosylation prevented archaellum assembly in *M. maripaludis*⁵⁷." It should be noted that similar results were obtained for individual point mutations in *Haloferax volcanii* (Tripepi et al 2012).

3. Fig 7: it should be stated more clearly that the depicted "possible interactions and potential clashes" are not based on molecular modelling or distance measurements, and that the placement of the two filaments is just used for illustrating the hypothesis, not based on experimental evidence. Furthermore, the filaments between the different strains still appear to be either not aligned vertically, or be shown in different rotational angles, since the glycosylation sites between the strains are not in vertical alignment.

Reviewer #2 (Remarks to the Author):

The authors have made an effort to address the concerns and the revised version of the manuscript has been improved according to the reviewers' recommendations. There are one major and two minor issues I noticed while reading the revised paper:

Major point:

(1) Regarding identifying the correct flagellin(s), I am not convinced by the result and the authors may have misunderstood my point. Sure, at residue 106/158, it's hard to tell. But how about residues 70-80 or residues 115-130 where the length of main chain backbone significantly varies between five candidates? If you can trace the full backbone, that should tell you something. As I said in the previous comment, if the author claims they cannot identify the flagellin(s), they need to show maps for ALL (not one) areas where those five sequences differ and let the readers see them. Note this is only a pure structure approach; how about comparing the gene cluster structure with other known archaeal flagellum?

Minor points:

(2) Map resolution (Å) FSC threshold 0.143 in Table S1. Please limit the estimated resolution to one-tenth of Ångstrom. (e.g. 3.23 should be 3.2) There is no way such estimation is meaningful at one-hundredth of Ångstrom.

(3) Remove all the B-factor rows from the table. Those numbers in EM are not meaningful. What's worse is that the unit used is Å² and table include all negative numbers...

Reviewer #3 (Remarks to the Author):

review:

Mashni et al. have responded well to all of my comments. I appreciate the sincerity with which they approached revisions and naturally feel the work is much improved.

I agree that it is not worth pursuing very difficult experiments with quantum dot labelling of filaments, and approve wholeheartedly of the motility data now present in Fig. 5 and in SI, and the revisions in the writing to aid at least my understanding of the work (and I believe also other readers).

The primary revision is the addition of the tracking measurements, Fig. 5 in main text and the SI movie which shows clearly phenotypic changes according to the genotype.

I also appreciate the consideration of my suggestions to improve figure clarity and some uncertainties I had on my reading of the work.

I find Fig 2. much clearer now also, the coloration of a single archaellin is helpful to understand the areas in which I was confused previously with the two helices mentioned.

I recommend this paper for publication.

We thank the reviewers for their constructive feedback and for appreciating our efforts to address their comments. We are pleased that all reviewers think that the manuscript has improved. We provide a revised manuscript that includes text additions and edits marked in red and two revised figures (Fig. 7 and Extended Data Fig. 4). We further provide below a point-by-point response to the reviewers' comments, with a description of the changes incorporated into the manuscript.

Reviewer #1 (Remarks to the Author):

The revised manuscript with the updated title “Perturbed N-glycosylation of *Halobacterium salinarum* archaeellum filaments leads to filament bundling and compromised cell motility” by Sofer et al. is a substantially improved version of the original manuscript, and in addition to the detailed structural analysis of *Halobacterium salinarum* archaeella now presents data on the motility of individual cells. The manuscript focuses on the impact of impaired N-glycosylation on both archaeella structure and resulting effects on cell motility. This revision includes additional details on the already commendable structural analysis of the archaeellum filaments. In addition, the new motility analyses of individual cells provide valuable insights into the swimming behavior of cells with perturbed N-glycosylation. Unfortunately, despite the additional data, the functional insights are still very limited.

Major Comments:

1. In their response the authors stated that “earlier efforts attributed the loss of motility to an inability or compromised ability to assemble archaeellum filaments by the different N glycosylation mutants.”, and in the manuscript similar statements (e.g. “The explanation offered for these observations was that compromised N-glycosylation affects archaeellum assembly and structure, and by extension, function, reflected as compromised swimming ability.”, and “As such, the N-linked glycans on the surface of archaeellins were assigned roles in promoting archaeellum filament assembly and stability, which in turn, facilitates motility”) are made. The authors then contrast these previous findings with their analysis of archaeella structure and cell motility for strains lacking the last two sugars of the N-linked tetrasaccharide in *Hbt. salinarum* to highlight that archaeellum filament assembly or structure was not significantly affected. However, this comparison is a misleading representation of previous findings. Previous studies not only in *Hbt. salinarum*, but also in *Hfx. volcanii*, *Methanococcus voltae*, and *Methanococcus maripaludis* indicated that motility is only fully lost in some N-glycosylation pathway mutants (e.g. if the OST AglB is deleted, or enzymes involved in the attachment of the first monosaccharides) while other N-glycosylation pathway mutants (often for enzymes involved in the attachment of the last monosaccharide(s)) exhibited reduced motility, implying that a functional archaeellum is still assembled. The manuscript presented here confirms these findings on the structural level, but the distinction between motile and non-motile N-glycosylation pathway mutants, and their corresponding implications on archaeellum assembly need to be stated more clearly in both the introduction and discussion, to make clear that previous studies already showed that not all monosaccharides are needed for archaeellum assembly, and that the results of the current study do not contradict previous findings about the disrupted archaeellum assembly e.g. in *aglB* mutants.

Prior this report, earlier efforts had assigned the impact of truncated N-glycosylation on cell motility to perturbed archaeellum assembly (even though most did not actually examine the archaeella generated in these mutants). As such, the Introduction refers to these previous ideas. Over the course of the present report, we introduce an additional consideration for

why compromised N-glycosylation hinders motility, namely, archaeellum filament bundling. At no point do we discount the earlier results. Indeed, we specifically state in the Discussion that “we cannot rule out that the first and second sugars are somehow required for archaeellin folding, stabilization and/or packing. Indeed, deletion of oligosaccharyltransferase-encoding *aglB* not only abolished archaeellin N-glycosylation but also resulted in a complete loss of archaeella in *Hbt. salinarum*²⁸, as well as in *Methanococcus voltae*³¹, *Methanococcus maripaludis*⁵⁶, and *Haloferax volcanii*³⁴. Likewise, deletion of genes encoding glycosyltransferases that add the first and/or second N-linked glycan sugars in these species also prevented archaeellum assembly^{29,30,32,34,56}. Moreover, mutating all four archaeellin asparagine residues destined for glycosylation prevented archaeellum assembly in *M. maripaludis*⁵⁷. In conclusion, while the requirements of N-glycosylation for archaeellum filament assembly and stability are not entirely understood, it is clear that the two terminal sugars of the N-linked tetrasaccharide decorating *Hbt. salinarum* archaeellins are not essential for archaeellation or archaeellum filament assembly.” Furthermore, we end the Discussion with the question: “Why does the absence of N-glycosylation or the addition of only a single N-linked sugar prevent archaeellin filamentation?”

Still, in light of the reviewer’s comments, we qualified the text they noted, which includes more general statements. Specifically, the statement in the Introduction now reads: “Indeed, a similar effect of N-linked glycan truncation, or the absence thereof, has been seen in other archaea, **with the extent of compromised motility differing among different mutants and species**^{31–35}. The explanation offered **in some of these studies** was that compromised N-glycosylation affects archaeellum assembly and structure, and by extension, function, reflected as compromised swimming ability.” The statement in the Discussion now reads: “The truncation or absence of N-linked glycans that decorate archaeellins often results in compromised swimming^{28–35}. **Thus, the conclusion from some of these studies was that the N-linked glycans (or certain sugars comprising these structures) on the surface of archaeellins promote** archaeellum filament assembly and stability, which in turn, facilitates motility.”

2. The two main findings that are truly novel are the bundling of archaeellum filaments, and the altered single cell swimming behavior. However for both of these findings, the mechanistic reasons remain to be elucidated, and potential explanations that would represent alternatives to the suggested electrostatic repulsion or spatial hindrance are inappropriately discussed (see subsequent comments). As such, the functional or mechanistic insights provided by this study are rather limited. The study reveals filament bundling, but cannot clarify why or how this bundling occurs. Furthermore, the study shows that previous findings on the reduced motility of N-glycosylation pathway mutants is due to altered swimming behavior, but without elucidating the movement of the corresponding (potentially bundled) archaeella.

Regarding the mechanism of bundling, in the Discussion, we considered possible reasons why glycosylation by the complete N-linked tetrasaccharide limits archaeellum filament bundling, namely, steric hinderance and/or electrostatic considerations, and by extension, considered why prevention of bundling would not occur in the mutants considered. In response to a similar comment that the reviewer made earlier, we analyzed filament pairs and demonstrated in Extended Data Fig 10b and corresponding data in the Results section (p. 20) that the interactions between filaments are not specific and mentioned that they are unlikely to be governed by electrostatic interactions (due to the presence of 4 M NaCl in our samples), but rather by hydrophobic sticking. We now added to the Discussion the following

statement: "Still, detailed understanding of how filament bundling occurs will require further investigation."

To elucidate the movement of individual archaellum filaments within a cell with altered swimming behavior, we would need to label archaellins of mutant strains with quantum dots and perform advanced imaging and analyses (as reported in Kinoshita et al., *Nature Microbiology* 2016). We agree that this kind of work would directly demonstrate the connection between altered swimming and filament bundling, but it will also take a significant amount of time, is not worth pursuing for this study, and would justify a stand-alone publication, as we explained previously. Indeed, reviewer #3 agreed with us on this matter ("I agree that it is not worth pursuing very difficult experiments with quantum dot labelling of filaments, and approve wholeheartedly of the motility data now present in Fig. 5 and in SI"). We now added to the Discussion the following statement: "Future single-cell analyses of archaellum filaments during swimming⁵⁰ will be useful to establish a direct connection between filament bundling and perturbed motility in these mutant strains."

We believe that these two main findings (in addition to other findings not mentioned here), which the reviewer describes as novel, merit publication.

3. Results on the number of archaellum filaments in the different strains are presented in a confusing way. In their comments, the authors state that their Fig 6 "misled the reviewer into concluding that the mutant strains produced higher levels of archaellum filaments than did the parent strain". However, the current data still suggest said higher levels of archaellum filaments in the mutant strains. Based on the authors source data for Fig 6, a significant difference in the number of archaella per cell was observed between the WT (an average of 1.6 filaments per cell) and the *agl27* mutant strain (an average of 2 filaments per cell). It is unclear why this increase in filaments per cell would not potentially contribute to the observed increase in bundling. The authors also state that the mutants show a decreased shearing of archaella during sample preparation (or during growth), but it is unclear why the perturbed N-glycosylation (and/or the bundling) would lead to an altered shearing behavior. Finally, the authors also state that "It is unlikely that the enhanced filament bundling seen with the mutant strains is due to increased archaellin synthesis, relative to the parent strain, given our earlier qRT-PCR results, which showed that the transcription of *Hbt. salinarum* archaellins was either unaltered or reduced in Δ *agl27* and Δ *agl26* cells". However, these previous results showed unaltered transcription for 2-3 of the 5 tested archaellins, and the corresponding protein abundances have not been tested (e.g. via quantitative mass spectrometry). A difference in protein abundance therefore cannot be ruled out. Furthermore, since the identity of the individual subunits in the filament could not be revealed, it is possible that the ratio of different archaellins in the filament changed and thereby affected the observed curvature and bundling.

The reviewer argues for the case that the observed bundling is the result of augmented archaellum filament levels in the mutant strains. In our previous response to the reviewers, in which we wrote: "misled the reviewer into concluding that the mutant strains produced higher levels of archaellum filaments than did the parent strain", we referred to panels we had provided for Fig. 6a. We subsequently revised the figure to present comparable numbers of filaments in panels for each strain to demonstrate that bundling was observed on the cryo-EM grid only for the mutant. Once this was established, a second observation was that, on average, there were indeed more archaellum filaments found on *agl27* mutant strain cells than on parent strain cells, as we explicitly report in the text (by extension, this indicates decreased shearing of archaella in the mutant). In claiming that the mutant strain produces

more archaellum filaments, which would explain the observed bundling, the reviewer argues that the difference in average number of archaellum filaments per cell is significant. First, it is not clear how the reviewer determined this difference to be significant. Second, it seems unlikely that such an increase from 1.6 to 2 (if indeed significant) would result in the massive bundling observed. Indeed, we did not find dramatically larger numbers of archaellum filaments in mutant strain cells; in both strains the number of archaella per cell ranged from 0 to 4 (see Source Data). Archaellum filaments were found twisted around each other in the *agl27* mutant even when only two filaments were observed on the cell.

Regarding our earlier transcription data, the reviewer is correct in that the levels of some archaellin transcripts were decreased in the N-glycosylation mutants, while some were not, relative to the levels detected in the parent strain. Clearly, however, no increases in total transcript levels of the archaellins were observed, which argues against increased archaellin levels (this information was already added to the Discussion). As such, there is no real justification to confirm this at the protein level by MS.

The reviewer raises the possibility that our glycosylation mutants assemble filaments using different combinations of archaellins than does the parent strain. Although others have addressed the impact of modifying the composition of archaella by deleting genes encoding specific archaellins in *Hbt. salinarum* (Tarasov et al., 2000) and other archaea (e.g., Pyatibratov et al., 2008; Tripepi et al., 2010; Tripepi et al., 2012; Pyatibratov et al., 2020), none reported bundling as seen here. However, similar bundling of archaellar pili was observed in a mutant with perturbed glycosylation in an earlier study (as cited in the Discussion). As such, the most reasonable explanation for the bundling observed in the glycosylation pathway mutants is perturbed glycosylation.

Still, in light of the reviewer's comments we added to the Discussion the following statement: "However, we cannot rule out completely that bundling in the N-glycosylation mutant strains is also affected by changes in the relative expression of the various archaellins."

4. A direct connection between the altered swimming behavior and the bundling of filaments is implied but not confirmed through experimental evidence. It is unclear whether the observed swimming cells in the mutant have bundled archaella, or whether they are examples of cells with single/separated archaella. This distinction would be important to conclude whether the bundling itself is linked to the altered swimming behavior, or whether additional factors play a role.

We demonstrated that archaellin filaments lacking the last and the last two sugars of the N-linked tetrasaccharide decorating these assemblies bundle and such bundling increases as the N-linked glycan shortens from four to two sugars. At the same time, swimming behavior was seriously restricted in distance and direction in direct correlation to the extent of bundling and the degree of glycan shortening. All of these observations are novel, alone meriting publication. As noted above, we agree that it would be ideal to demonstrate a direct connection between altered swimming and archaellum filament bundling but labeling the archaella to be visible during swimming and the associated imaging and analyses are well beyond the scope of this manuscript and would merit a stand-alone publication.

Minor comments:

1. "Likewise, deletion of genes encoding the glycosyltransferases adding the first and second N-linked glycan sugars in *Hbt. salinarum*29,30 and *M. voltae*32 also prevented

archaellum assembly.” It should be noted that the same was also observed for *Methanococcus maripaludis* (VanDyke et al 2009) and *Haloferax volcanii* (Tripepi et al 2012)

These references have been added to the revised text.

2. “Moreover, mutating all four archaellin asparagine residues destined for glycosylation prevented archaellum assembly in *M. maripaludis*⁵⁷.” It should be noted that similar results were obtained for individual point mutations in *Haloferax volcanii* (Tripepi et al 2012).

In the *Hfx. volcanii* study, while it is true that the three modified Asn residues were mutated, only motility analyses and SDS-PAGE migration results were presented (see Fig 8 in that study), rather than results that describe archaella assembly. In the study we cite on *M. maripaludis*, the assembly of archaella was directly assessed using negative-stain EM of intact cells. Such analysis was not performed in the *Hfx. volcanii* study and thus this reference is irrelevant here, where we discuss filament assembly.

3. Fig 7: it should be stated more clearly that the depicted “possible interactions and potential clashes” are not based on molecular modelling or distance measurements, and that the placement of the two filaments is just used for illustrating the hypothesis, not based on experimental evidence. Furthermore, the filaments between the different strains still appear to be either not aligned vertically, or be shown in different rotational angles, since the glycosylation sites between the strains are not in vertical alignment.

In light of the reviewer’s comment, we added the word “schematic” to the figure legend and mentioned that the maps were placed manually side by side to illustrate possible interactions between the filaments. Moreover, we have modified the figure such that all glycosylation sites are vertically aligned.

Reviewer #2 (Remarks to the Author):

The authors have made an effort to address the concerns and the revised version of the manuscript has been improved according to the reviewers' recommendations. There are one major and two minor issues I noticed while reading the revised paper:

Major point:

(1) Regarding identifying the correct flagellin(s), I am not convinced by the result and the authors may have misunderstood my point. Sure, at residue 106/158, it's hard to tell. But how about residues 70-80 or residues 115-130 where the length of main chain backbone significantly varies between five candidates? If you can trace the full backbone, that should tell you something. As I said in the previous comment, if the author claims they cannot identify the flagellin(s), they need to show maps for ALL (not one) areas where those five sequences differ and let the readers see them. Note this is only a pure structure approach; how about comparing the gene cluster structure with other known archaeal flagellum?

We apologize for the misunderstanding. New Extended Data Fig. 4 contains panels of all regions that differ across archaellins, showing sequence alignment, cryo-EM maps, and various models of archaellins fit into these maps (the only region not shown is residue 42 in the conserved α helix, which corresponds to Phe in ArlA1/2 and to Tyr in ArlB1/2/3, a difference that cannot be resolved at this resolution). While in some regions certain models fit the density better than others, none of the models fit the density well in all regions, leading to ambiguity in determining the identity of the archaellins in each position within the filament.

As far as we understand, the reviewer suggests comparing the gene cluster structure with those of other known archaeella to figure out which of the five archaeellins are expressed and comprise the archaeellum filament. In *Hbt. salinarum*, the two *arlA* genes (*arlA1* and *arlA2*) are found in tandem on the chromosome, distant from the position of the three *arlB* genes (*arlB1*, *arlB2*, and *arlB3*), which are also found in tandem (see figure below). All five archaeellins were detected in preparations of isolated archaeella (i.e., spent growth medium), by SDS-PAGE and mass spectroscopy (Gerl et al., 1989 and many subsequent reports). We revised the text in the Results to explain that the five archaeellins are expressed, rather than just encoded. Several studies of specific archaeellin deletion mutants indicated that both ArlA and ArlB archaeellins are needed for proper archaeellum assembly and function (Tarasov et al., 2000, Syutkin et al., 2012, Beznosov et al., 2007). Tarasov et al.(2000) proposed that the region proximal to the cell body is composed of ArlB archaeellins, whereas the rest of the filament comprises ArlA archaeellins. Our cryo-EM map suggests that our samples contain a mixture of at least ArlA2 and other archaeellins (see Extended Data Fig. 8), and as shown in new Extended Data Fig. 4, it is difficult to resolve the identity of those other archaeellins, and more importantly, their distribution along the filament. To our knowledge, *Hbt. salinarum* is the most studied species with such a gene cluster structure. Unfortunately, we cannot infer the archaeellin distribution in the *Hbt. salinarum* archaeellum filament from other species with a known archaeellum composition, due to the differences in gene organization and expression.

Gene cluster structure of *Hbt. salinarum*. From Jarrell et al., 2021.

Taking together the lack of resemblance of gene clusters to species with known archaeellum composition, the experimental evidence of multiple archaeellins comprising the archaeellum in *Hbt. salinarum*, and the ambiguity in the cryo-EM map, we cannot determine the distribution of archaeellins within the filament at this time. We think that further high-resolution structural work on mutants lacking certain archaeellin genes would be required to do so.

Minor points:

(2) Map resolution (\AA) FSC threshold 0.143 in Table S1. Please limit the estimated resolution to one-tenth of \AA ngstrom. (e.g. 3.23 should be 3.2) There is no way such estimation is meaningful at one-hundredth of \AA ngstrom.

The resolutions listed in Table S1 have been changed to estimations limited to one-tenth of an \AA .

(3) Remove all the B-factor rows from the table. Those numbers in EM are not meaningful. What's worse is that the unit used is \AA^2 and table include all negative numbers...

The B-factor rows were removed from the table.

Reviewer #3 (Remarks to the Author):

Mashni et al. have responded well to all of my comments. I appreciate the sincerity with which they approached revisions and naturally feel the work is much improved.

I agree that it is not worth pursuing very difficult experiments with quantum dot labelling of filaments, and approve wholeheartedly of the motility data now present in Fig. 5 and in SI,

and the revisions in the writing to aid at least my understanding of the work (and I believe also other readers).

The primary revision is the addition of the tracking measurements, Fig. 5 in main text and the SI movie which shows clearly phenotypic changes according to the genotype.

I also appreciate the consideration of my suggestions to improve figure clarity and some uncertainties I had on my reading of the work.

I find Fig 2. much clearer now also, the coloration of a single archaellin is helpful to understand the areas in which I was confused previously with the two helices mentioned.

I recommend this paper for publication.

No revisions requested.

Reviewer #2 (Remarks to the Author):

The authors have addressed my comments, and I have no further concerns.